# Repair of mismatched templates during Rad51-dependent Break-Induced Replication

**Jihyun Choi[1], Muwen Kong[2], Danielle N. Gallagher[1], Kevin Li[3], Gabriel Bronk[3], Yiting Cao[1], Eric C. Greene[2], James E. Haber[1]***

1 Department of Biology and Rosenstiel Basic Medical Sciences Research Center, Brandeis University, Waltham, Massachusetts, United States of America, 2 Department of Biochemistry & Molecular Biophysics, Columbia University, New York, New York, United States of America, 3 Department of Physics, Brandeis University, Waltham, Massachusetts, United States of America

* haber@brandeis.edu

**Data Availability Statement:** All relevant data are within the manuscript and its Supporting Information files.

**Funding:** This work has been supported by the National Institute of Health grant R35GM127029 to

## Abstract

Using budding yeast, we have studied Rad51-dependent break-induced replication (BIR), where the invading 3' end of a site-specific double-strand break (DSB) and a donor template share 108 bp of homology that can be easily altered. BIR still occurs about 10% as often when every 6th base is mismatched as with a perfectly matched donor. Here we explore the tolerance of mismatches in more detail, by examining donor templates that each carry 10 mismatches, each with different spatial arrangements. Although 2 of the 6 arrangements we tested were nearly as efficient as the evenly-spaced reference, 4 were significantly less efficient. A donor with all 10 mismatches clustered at the 3' invading end of the DSB was not impaired compared to arrangements where mismatches were clustered at the 5' end. Our data suggest that the efficiency of strand invasion is principally dictated by thermodynamic considerations, i.e., by the total number of base pairs that can be formed; but mismatch position-specific effects are also important. We also addressed an apparent difference between *in vitro* and *in vivo* strand exchange assays, where *in vitro* studies had suggested that at a single contiguous stretch of 8 consecutive bases was needed to be paired for stable strand pairing, while *in vivo* assays using 108-bp substrates found significant recombination even when every 6th base was mismatched. Now, using substrates of either 90 or 108 nt–the latter being the size of the *in vivo* templates–we find that *in vitro* D-loop results are very similar to the *in vivo* results. However, there are still notable differences between *in vivo* and *in vitro* assays that are especially evident with unevenly-distributed mismatches. Mismatches in the donor template are incorporated into the BIR product in a strongly polar fashion up to ~40 nucleotides from the 3' end. Mismatch incorporation depends on the 3'→ 5' proofreading exonuclease activity of DNA polymerase δ, with little contribution from Msh2/Mlh1 mismatch repair proteins, or from Rad1-Rad10 flap nuclease or the Mph1 helicase. Surprisingly, the probability of a mismatch 27 nt from the 3' end being replaced by donor sequence was the same whether the preceding 26 nucleotides were mismatched every 6th base or fully homologous. These data suggest that DNA polymerase δ "chews back" the 3' end of the invading strand without any mismatch-dependent cues from the strand invasion structure. However, there appears to be an alternative way to incorporate a mismatch at the first base at the 3' end of the donor.

J.E.H. D.N.G was supported by NIGMS Genetic Training Grant T32GM007122 (https://www.nigms.nih.gov/) and by the National Science Foundation Graduate Research Fellowship Program under grant 1744555 (https://nsfgrfp.org/). E.C.G was supported by the National Institute of Health grant R35GM118026. The funders had no role in study design, data collection and analysis, decision to publish, or preparation of the manuscript.

**Competing interests:** The authors have declared that no competing interests exist.

## Author summary

DNA double-strand breaks (DSBs) are the most lethal forms of DNA damage and inaccurate repair of these breaks presents a serious threat to genomic integrity and cell viability. Break-induced replication (BIR) is a homologous recombination pathway that results in a nonreciprocal translocation of chromosome ends. We used budding yeast *Saccharomyces cerevisiae* to investigate Rad51-mediated BIR, where the invading 3' end of the DSB and a donor template share 108 bp of homology. We examined the tolerance of differently distributed mismatches on a homologous donor template. A donor with all 10 mismatches clustered every 6th base at the 3' invading end of the DSB was not impaired compared to arrangements where mismatches were clustered at the 5' end. We also compared the efficiency of *in vivo* BIR with *in vitro* D-loop formation and find that for substrates of the same length, the tolerance for mismatches is comparable. However, there are still notable differences between *in vivo* and *in vitro* assays that are especially evident in substrates with unevenly-distributed mismatches. Mismatches are incorporated into the BIR product in a strongly polar fashion as far as about 40 nucleotides from the 3' end, dependent on the 5' to 3' proofreading activity of DNA polymerase δ. Pol δ can "chew back" the 3' end of the invading strand even when the sequences removed have no mismatches for the first 26 nucleotides. However, a mismatch at the first base can be removed from the 3' end by another, unidentified mechanism.

## Introduction

DNA double-strand breaks (DSBs) are the most toxic lesions that can occur in DNA, and failure to repair these breaks can result in genome instability. Eukaryotes have evolved two major types of DNA repair mechanisms to deal with DSBs: non-homologous end joining (NHEJ) and homologous recombination (HR). In "classic" NHEJ, broken ends are ligated back together using little or no base pairing of the broken ends whereas microhomology-mediated end-joining (MMEJ), also known as alternative nonhomologous end-joining (Alt-NHEJ) uses several bases of shared microhomology at the junction to align the broken strands that are revealed after DSB ends are resected by 5' to 3' exonucleases [1–4]. Both NHEJ and MMEJ may result in small insertions or deletions of sequences surrounding the DSB.

When both ends of a DSB share homology with a donor–on a sister chromatid, a homologous chromosome or at an ectopic location–repair most often occurs by gene conversion in which the DSB is patched up by copying the intact donor sequence [5]. HR is initiated by 5' to 3' resection that creates 3'-ended single strands that are bound by the Rad51 recombinase that facilitates the search for homology and promotes strand invasion, enabling the 3' end of the invading strand to be extended by DNA polymerase. In synthesis-dependent strand annealing, the most common gene conversion outcome in mitotic cells, subsequent steps allow the second end of the DSB to anneal to the newly copied strand and initiate another round of DNA synthesis to complete the repair.

Here we focus on break-induced replication (BIR), an alternative HR pathway that is engaged when only one of the ends of a DSB shares homology with a donor sequence [5]. BIR allows the extension of eroded telomeres and the re-initiation of DNA replication at stalled and broken replication forks [6,7]. As in gene conversion, BIR is initiated by 5'→ 3' resection of a broken DNA end to generate a 3' single-stranded DNA (ssDNA) tail (S1A Fig). Initially, replication protein complex A (RPA) coats the ssDNA tails but is then displaced by the

recombination protein Rad51 [8,9]. Each monomer of Rad51, like its bacterial homolog, RecA, binds 3 nucleotides of ssDNA to form a nucleoprotein filament that catalyzes base-pairing and strand invasion between the Rad51-coated ssDNA end of DSB and a homologous double-stranded DNA (dsDNA) donor [10–13]. Strand invasion and the formation of a displacement loop (D-loop) enables DNA polymerase δ to prime DNA synthesis and extend the 3' end of invading strand to the end of the chromosome [14–16]. In many respects, BIR events studied in budding yeast resemble mammalian alternative lengthening of telomeres (ALT) [7,17–19]. BIR also appears to be critical for DNA synthesis that occurs very late in the cell cycle as cells enter mitosis (MiDAS) [7,20–22]. Both in yeast and in mammalian cells, the nonessential Pol32 (POLD3) subunit of DNA polymerase δ is required for BIR but not for normal DNA replication [23,24].

Precisely how Rad51 performs strand exchange remains a subject of great interest [25–28]. HR depends on both the length and the degree of homology between donor and recipient. In budding yeast, efficient gene conversion or BIR can be accomplished with at least 50–100 bp of sequence homology [17,27,29]. In the intrachromosomal BIR system that we use in this study (Fig 1), a site-specific DSB is induced by the expression of HO endonuclease and repair occurs at an ectopic 108-bp donor sequence. Approximately 14% of cells repaired the DSB by BIR when the homologous region is 108 bp; but repair dropped to 1% with a 54-bp template and to 0.02% when there were only 26 bp [17]. Repair was also reduced by the presence of heterologies (the presence of non-complementary bases between donor and recipient sequences). However, repair was still possible when every $6^{th}$ base was mismatched, at a rate that was about 10% of that seen when the templates were identical [17]. These *in vivo* results contrasted with *in vitro* single-molecule studies using substrates with just a short tract of sequence homology, flanked by regions without sequence homology, which found that a minimum of at least 8 consecutive bases must be present for a stable association with double-stranded DNA promoted by the bacterial strand exchange protein RecA, or its eukaryotic homologs Rad51 or Dmc1 [10]. Other *in vitro* studies also concluded that 8 contiguous base pairs are required for stable substrate binding by bacterial RecA [30–32]. These differences could have reflected the fact that some key recombination proteins acting *in vivo* were not included in the *in vitro* assays; but in fact, as we show here, the differences among these studies likely reflect the differences in the overall size of the homologous regions that were studied. When we performed *in vitro* D-loop assays mediated by Rad51 in the presence of Rad54, analyzing substrates identical to the 108-base regions used in our *in vivo* studies, the results are comparable, with D-loop formation occurring even when every $6^{th}$ base was mismatched.

In our previous study, we used donor templates in which the heterologies were evenly-distributed [17]. Here, we extended our study of mismatch tolerance by examining several templates each with ten mismatches but distributed unevenly, every $6^{th}$ base. We wished to determine if these different distributions would be treated equivalently, which would be the case if strand invasion in BIR were principally governed by thermodynamic considerations, in which the total number of base pairs that can be formed would dictate repair efficiency. Donor templates with majority of mismatches towards the 3' end proved to be nearly as efficient as a donor with 10 mismatches evenly-distributed (every 10 bases); but, unexpectedly, donor templates with majority of mismatches positioned towards the 5' end were statistically significantly lower than the evenly-distributed control. There were distinctive differences between the *in vivo* results and those obtained for the *in vitro* D-loop assay, suggesting that other repair factors may play a role when mismatches are closely spaced. However, this discrimination does not appear to be monitored by the Msh2-dependent mismatch repair system.

A second important aspect of our analysis of repair involving mismatched substrates came from analyzing the assimilation of heterologies from the donor into the BIR product.

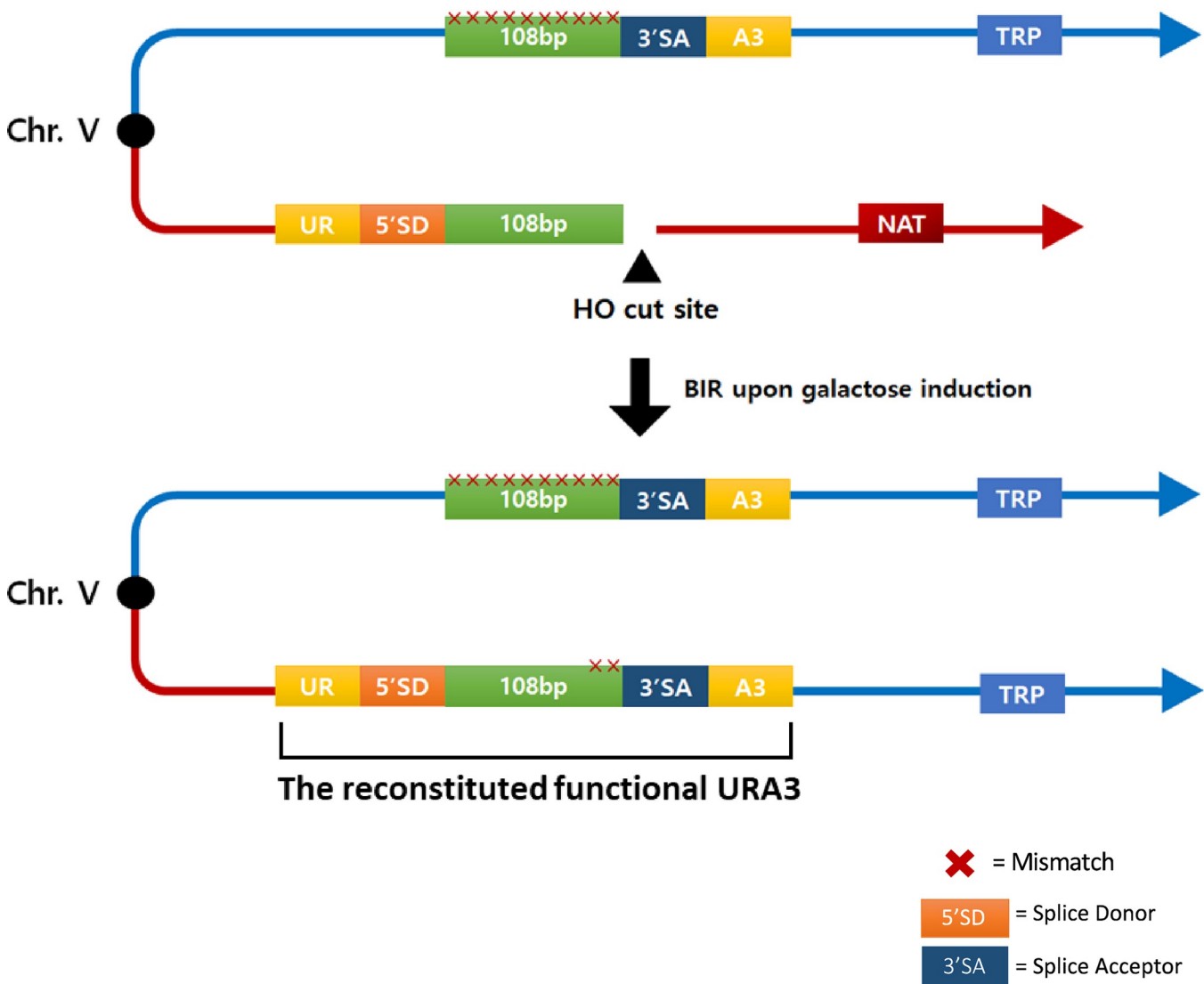

**Fig 1. BIR-dependent formation of a functional Ura3⁺ recombinant.** The recipient sequence shares the 108-bp region of homology and contain the 5'
sequences from the *URA3* gene (UR), the splice donor site (5' SD) of an artificial intron and the HO cut site. 108-bp donor sequences containing different
mismatch distributions were assembled into a plasmid containing the 3' sequences from the *URA3* gene (A3), the 3' splice-acceptor (3' SA) of the intron and
the *TRP1* auxotrophic maker. A DSB was created using the galactose-inducible HO endonuclease. This break is repaired by BIR using the donor sequences that
share 108-bp of homology located on the opposite arm of chromosome V. Once BIR is complete, a functional intron is formed, and yeast become Ura3⁺
recombinants.

Mismatches close to the 3' end of the invading strand are very frequently replaced by the template sequence (i.e. a gene conversion event), but there is a steep decline in their incorporation further away from the 3' end, so that by about 40 bp from the 3' end, incorporation of the template sequence is rare [17]. The assimilation of the template sequence is dependent on the 3' to 5' proofreading activity of DNA polymerase δ, which presumably chews back the 3' end of the invading strand before initiating copying of the donor template (S1B Fig). There was little effect when Msh2/Mlh1-dependent mismatch repair was ablated, although the extent of mismatch assimilation was shortened [17]. Here, we make the surprising discovery that 3' to 5' removal of the 3' end of the strand-invading DNA is evident even when the first 26 nucleotides are completely homologous to the template, suggesting that this resection is not provoked by a

nearby mismatch. Moreover, there appears to be another means of removing a heterology at the first base position.

## Results

To study the effect of different distributions of mismatches on BIR repair efficiency we used the assay shown in Fig 1. A galactose-inducible site-specific DSB is created by HO endonuclease at a site just distal to the 5' end of the *URA3* gene (UR) joined to an mRNA splicing donor sequence (SD) [17]. Only the centromere-proximal end of the DSB shares homology with the donor, which in this case is located on the opposite arm of the same chromosome, about 30 kb from the telomere. The donor consists of a 108-bp region of homology such that the DSB end is perfectly matched to the donor (i.e., there are no additional nonhomologous sequences at the 3' end). The 108-bp homologous segment is adjacent to 3' splice acceptor (SA) site, followed by the 3' end of the *URA3* gene (A3). Thus, BIR results in a nonreciprocal translocation producing an intron-containing, intact *URA3* gene so that cells can grow in the absence of uracil.

We confirmed that strain yRA280 (with every 10th base pair mismatched) and yRA321 (with every 6th base pair mismatched) had reduced, but significant levels of repair compared to yRA253 (a fully homologous donor) (Fig 2B). We note that the % BIR we report in this publication are lower than those previously reported by Anand *et al.* (2017), although results are proportionally very similar. We are not certain what methodological or environmental conditions have changed, but the results have been confirmed by several authors and the conclusions from the previous publication and this work are the same. We then created six new strains, each of which had 10 mismatches, but arranged so that they were clustered every 6 bases apart (Fig 2A). The mismatches included both transversion and transition mismatches (S2 Table).

When compared to yRA280, donor template B, with all mismatches clustered at the 3' invading end of the DSB (Fig 2C) was nearly as efficient as yRA280. In contrast, a substrate with a 48 bp of perfect homology at the 3' end (Fig 2C, donor template D) was statistically significantly less efficient than yRA280. Thus, although the 3' end must be synapsed with the donor to allow DNA polymerase δ to initiate new DNA synthesis at the 3' end, BIR efficiency was less impaired when the 3' invading end of the DSB was mismatched (donor template B) than when distal 5' end of homologous sequence was mismatched (donor template D). However, among all 6 templates there was no clear correlation between the location of the mismatches and their efficiency of usage (S4 Fig). Moreover, donor templates E and F, with 26 bp perfect homology at the 3' end performed worse than the four other substrates.

The variation among these templates cannot be attributed to differences in thermal stability of base-pairing in the 108-bp region as measured by the calculated melting temperature (Tm) between complementary 108-nt DNA strands (S2 Table and Fig 3A). Interestingly, the slopes of the linear regression lines were nearly identical for the evenly-distributed controls ($y = 0.053x - 3.74$; $R^2 = 0.98$) as for the 6 unevenly spaced cases ($y = 0.053x - 4.08$; $R^2 = 0.62$) but the unevenly-distributed series lie on a line shifted below the controls. Thus the templates with a higher GC content that can form a more stable heteroduplex–and have a higher Tm—with the invading 3'-end of the DSB yield an increased BIR efficiency. We note that the invading strand is always the same sequence.

### Comparison of *in vivo* and *in vitro* strand invasion assays

We next investigated how our *in vivo* results compare with *in vitro* assays of recombination, specifically strand invasion, by analyzing the ability of a single-stranded template to form a D-

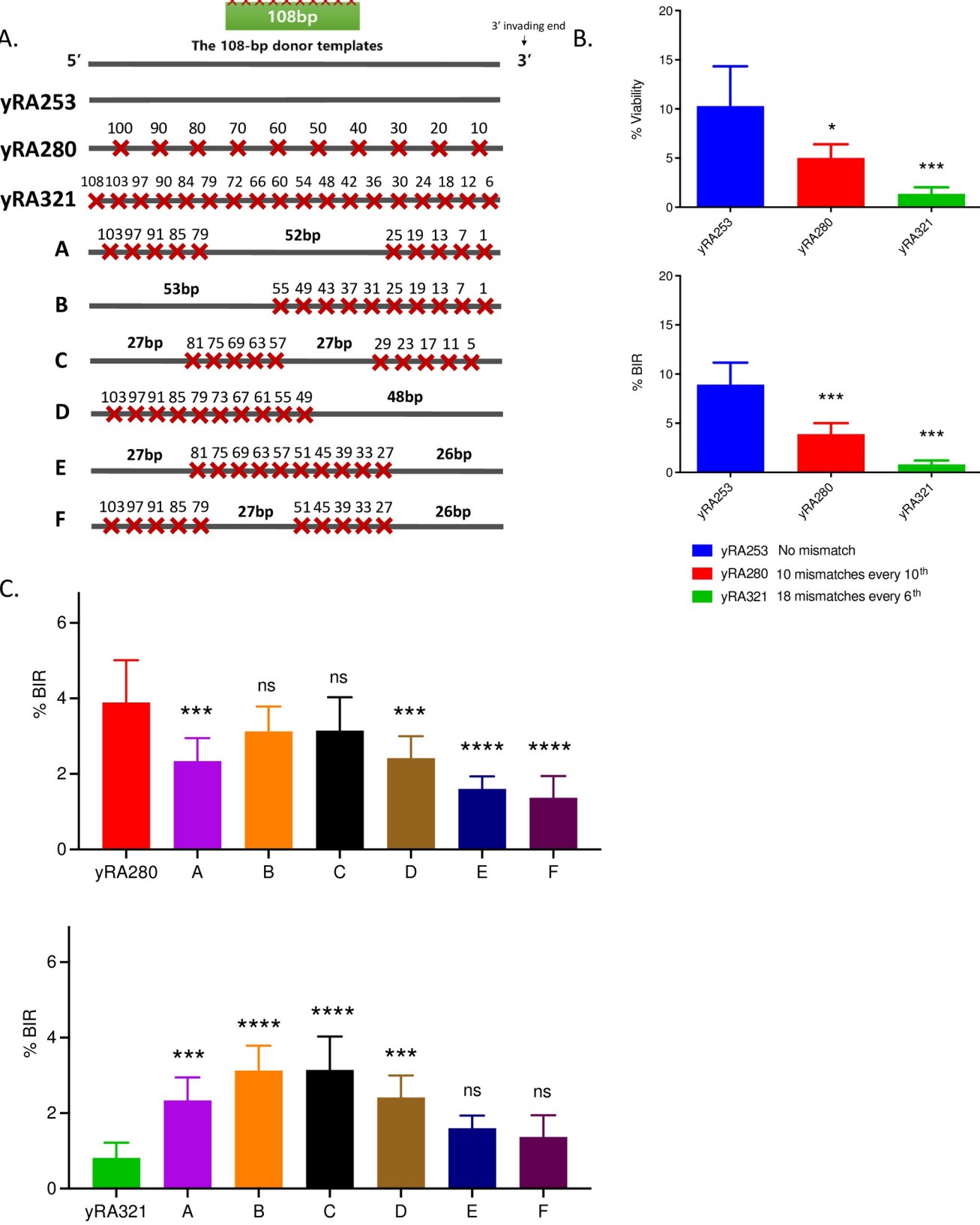

**Fig 2. The effect of different distributions of mismatches on a 108-bp donor template on the efficiency of Rad51-dependent BIR.** A. Arrangement of mismatches in 108-bp donors. yRA253 (no mismatch), yRA280 (10 mismatches every 10th bp) and yRA321 (18 mismatches every 6th). The set of donors with even mismatch distribution was compared with a set of divergent 108-bp donor templates containing a total of 10 mismatches that are distributed unevenly throughout the template. The spacing between the clustered mismatches are every 6th bp. DNA sequences are shown in S2 Table. B. Percent BIR efficiency of 108-bp donors with perfect homology and even mismatch distribution. One-way ANOVA multiple comparison was used to determine the p-value, $^{**}p \leq 0.001$, ns = not significant. Error bars indicate standard deviation. A minimum of three measurements were performed. C. Percent BIR efficiency of 108-bp donors with even and uneven distribution. The set of donors with uneven mismatch distribution was compared to yRA280 which contains the same mismatch density as all unevenly-mismatched donors and to yRA321 which has the same 6th bp spacing between clustered mismatches. Significance determined using a Dunnett's method (GraphPad Prism 9). Error bars refer to standard deviation. $^{**}$ p<0.001, ns = not significant.

loop with a supercoiled plasmid carrying a region of homology [14] (see Methods). Previous single molecule studies, using short duplex DNA of 70 base pairs in length, but designed to contain just a single tract of sequence homology $\geq 8$ bases in length had suggested that stable substrate binding required at a minimum of 8 consecutive base pairs of homologous sequence in assays with Rad51, Dmc1, or RecA, but lacking any other protein cofactors [33]. These data were quite different from the results found *in vivo*, where substrates containing 18 tracts of five consecutive base pairs of sequence homology (i.e. every 6th bp mismatched) had significant levels of recombination (Fig 2B). This difference could reflect the presence *in vivo* of additional protein factors not present in the single molecule *in vitro* assay (e.g., Rad52, Rad54, etc.), or differential substrate and protein requirements for stable substrate binding *in vitro* versus the complete recombination outcomes *in vivo*. Additionally, it could also be a consequence of the differences in the cumulative length of the homologous sequences within substrates themselves (i.e., 8 homologous base pairs *in vitro* versus a total of 90 homologous bases within the 108-bp substrate with every 6th base mismatched *in vivo*), where additional base pairing along a longer substrate might compensate for the lack of 8 consecutive paired bases. To investigate this question, we first carried out a series of *in vitro* D-loop assays with Rad51 and Rad54 using 90-base single-stranded DNA containing evenly-distributed mismatches (see Methods) (Fig 4A, S5 Table). Indeed, quite similar to the previous *in vivo* results (Fig 4B), D-loops could be detected even when every 6th base was mismatched. Overall, the influence of evenly-distributed mismatches on *in vivo* and *in vitro* assays proved to be strikingly similar (Fig 4B and 4C). Thus, the cumulative length of the homologous sequences present in these substrates allows mismatch densities as high as every 6th base pair to be tolerated during *in vitro* D-loop formation.

We then performed a similar D-loop analysis for the same six 108-nt unevenly-mismatched templates that we had analyzed *in vivo* (Fig 4D). As shown in Fig 3, there was a weak correlation between D-loop formation and Tm for both the *in vivo* (Fig 3A and 3B) and *in vitro* (Fig 3C) assays with these unevenly-distributed mismatches. But unlike the results *in vivo*, there was no general reduction in product formation compared to the evenly-distributed controls in the *in vitro* assays. Since the same set of donor template and ssDNA sequences were used in both *in vivo* and *in vitro* experiments, we also compared *in vivo* BIR efficiencies with the quantified *in vitro* D-loop product formation, which revealed a general positive correlation between these results (Pearson $R^2 = 0.43$, Fig 4E). A Mann-Whitney test confirmed that the results between these *in vivo* and *in vitro* sets are statistically significantly different (p = 0.001). There may be some additional factors *in vivo* such as mismatch-stimulated heteroduplex rejection [34,35] that reduce the likelihood of successful BIR. The variation among the *in vitro* D-loop outcomes also suggest there are likely to be additional sequence-specific features that have yet to be understood.

## Effect of the mismatch repair gene *MSH2* on BIR with mismatched templates

We then asked if the generally lower values for the unevenly-distributed series for BIR was attributable to the action of the mismatch repair protein, Msh2. Although Msh2 plays an

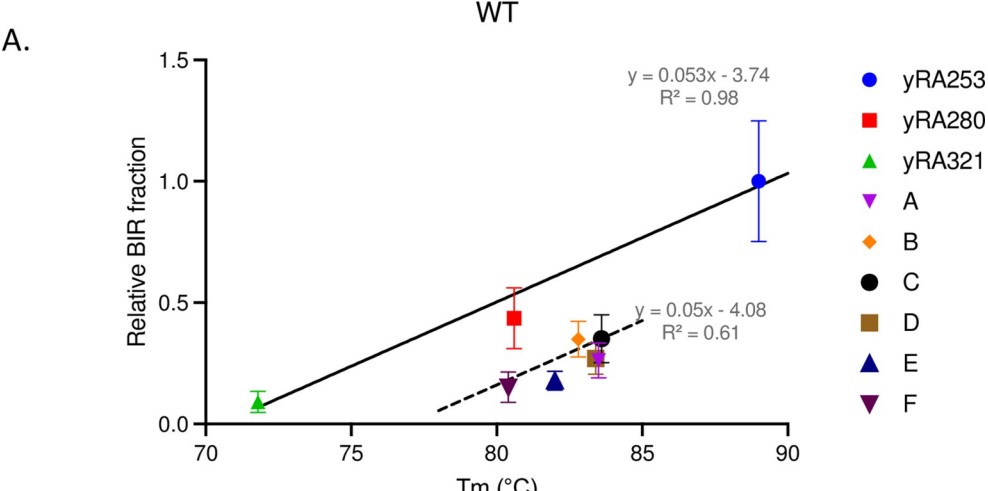

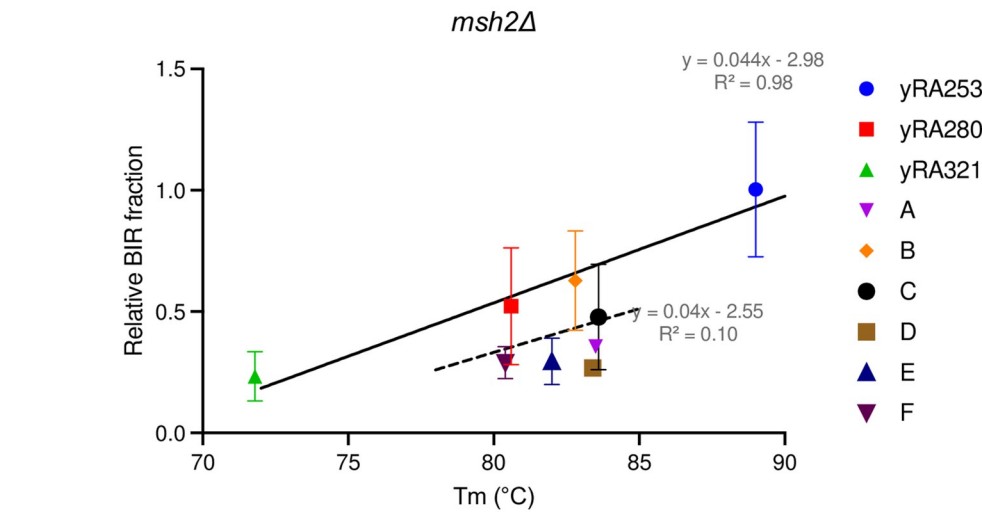

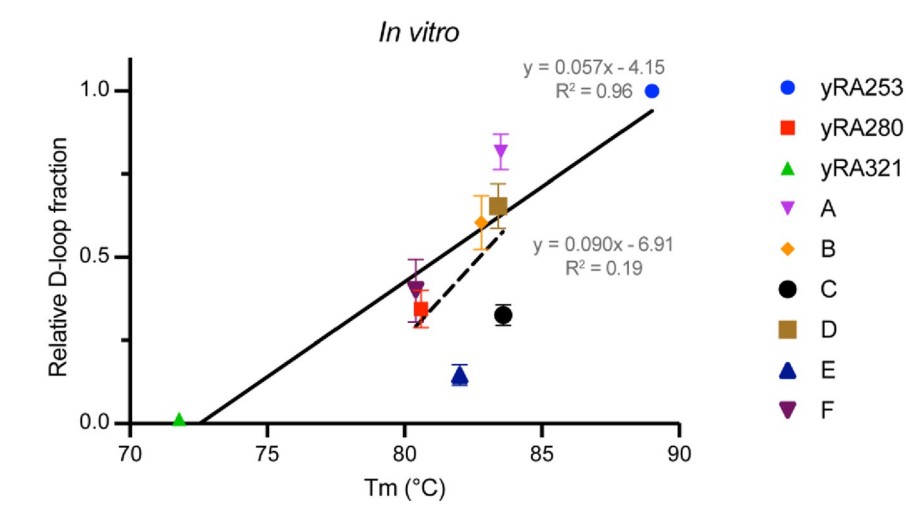

**Fig 3. Rad51-mediated strand annealing is less efficient when the mismatches are spaced every 6bp apart.** A. Percent BIR of both evenly- and unevenly-mismatched donor templates are plotted against their melting temperatures Tm (˚C). The melting temperature of single-stranded DNA of each donor template, synapsed with the complementary single-strand DNA of recipient sequence (S2 Table), was calculated by the method of Markham and Zucker [37]. A least-squares line (solid black line) was determined for the three samples with 0, evenly-distributed every 10th and evenly-distributed every 6th bp mismatches (yRA253, yRA280, and yRA321). A least-squares line for the six donor templates with 10 uneven mismatch distributions was plotted separately (black dotted line). B. Similar comparisons were made for strains deleted for *MSH2*. C. Efficiency of *in vitro* D-loop formation for 108-bp templates identical to A through F, as noted above. See also Fig 4.

important role in heteroduplex rejection, it proved to be primarily dependent on the presence of a nonhomologous tail at the 3' end of the invading strand [17]. Here there 3' end is perfectly matched. We assessed the effect of deleting *MSH2* on BIR efficiencies among the set of different donors. In general, the efficiency of BIR was lower in the absence of Msh2, even in the absence of mismatches in the donor (Fig 3). Compared to wild type cells, in the absence of Msh2 there was a lower slope of the relation between BIR efficiency and the Tm of the sequence, both for the three controls ($y = 0.044x - 2.98$ compared to $y = 0.053x - 3.74$) and for the 6 unevenly-distributed cases ($y = 0.036x - 2.55$ versus $y = 0.053x - 4.08$) (Fig 3B). Again, the six mismatched cases appeared to lie on a separate line, with a slope similar to that for the evenly-distributed controls. Although none of the pairwise comparisons between WT and *msh2Δ* for individual templates were statistically significant (S5 Fig), the differences between the WT and *msh2Δ* data as a whole were significant by a Mann-Whitney test ($p = 0.004$). Because *msh2Δ* did not suppress the difference between the evenly-spaced and the clustered mismatch arrangements, we conclude that there is some other factor beyond the mismatch repair machinery that causes the unevenly-distributed mismatches to be less successful in BIR than the evenly-distributed controls. We note also that we did not find any difference in the thermal stability of possible secondary structures that could be formed by the different 108-nt regions, viewed as single-strand sequences (see Methods).

## Assimilation of mismatches from the donor into BIR products reveals the role of DNA polymerase δ

Once heteroduplex DNA forms by strand invasion, mismatch correction may lead to the incorporation of donor sequences into the BIR product. However, the assimilation of heterologies into the BIR product does not proceed through the general Msh2/Mlh1-dependent mismatch repair system [17], but instead is dependent on the 3' to 5' exonuclease activity of DNA polymerase δ, which chews away 3' end of the invading strand and then copies the donor sequence [36]. Thus, mismatches very close to the 3' end of the invading DNA are very frequently replaced by the donor allele. There is a steep drop in the incorporation of donor alleles, extending 40–50 nt from the DSB end, after which there was little or no incorporation of the mismatches (Fig 5E) [17].

To determine the extent of incorporating donor mismatches into the final product, we sequenced approximately 50 BIR products from each of the templates shown in Fig 2A. In each case, mismatches were incorporated into the BIR product with the same strong polarity seen with the evenly-distributed mismatches [17]. Mismatch assimilation was seen as far as 55 nucleotides from the 3' invading end (Fig 5A, donor template B). For template D, containing 10 mismatches clustered near the 5' end and with 48 bp perfect homology near the 3' end of the break, none of the 10 mismatches were incorporated (Fig 5B). However, for all the constructs in which there were mismatches in the first 48 bp, the pattern of incorporation was the same (Fig 5E). Surprisingly, in donor templates E and F (Fig 5C), the degree of incorporation of mismatches at positions 27, 33 and 39 bp from the invading end was indistinguishable from the correction of these same sites when all the mismatches are present at the 3' end (Fig 5A,

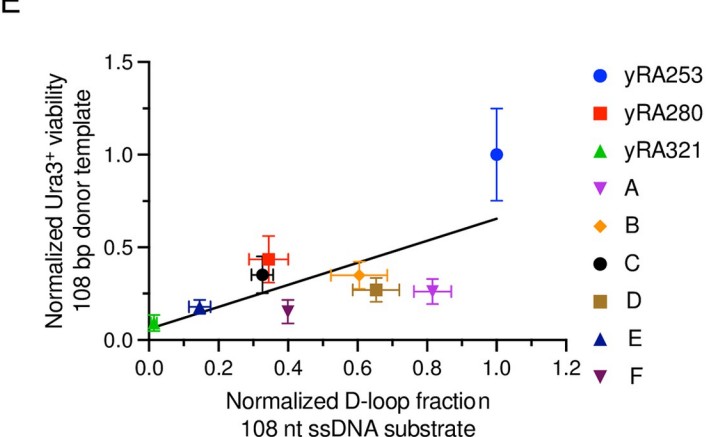

**Fig 4. Comparison of BIR with *in vitro* D-loop formation.** A. Normalized D-loop fraction, using 90-nt ssDNA substrates containing only evenly-distributed mismatches. B. Normalized Ura3+ viability (reproduced from [17]), using 108-bp donor DNA templates containing only evenly-distributed mismatches. C. Correlation between normalized Ura3+ viability and normalized D-loop fraction, using 108-bp donor templates and 90-nt ssDNA substrates, respectively, containing only evenly-distributed mismatches. D. Normalized D-loop fraction of 108-nt ssDNA substrates containing both evenly- and unevenly-distributed

mismatches (same sequences as those in Fig 2A). Statistical significance determined by comparing between donor templates A-F to yRA280 as control by using a Dunnett's method (GraphPad Prism 9). **** p<0.0001, ns = not significant. E. Comparison of normalized Ura3+ BIR efficiencies and D-loop fractions, using 108-bp donor templates and corresponding 108-nt ssDNA substrates, respectively, containing both evenly- and unevenly-distributed mismatches.

donor template B). These data suggest that 3' to 5' exonuclease activity of DNA polymerase δ removes the 3' end of the invading strand to incorporate mismatches beyond 26 bp even when this region lacks any mismatches.

Almost all mismatch incorporation was eliminated when the 3' to 5' proofreading exonuclease activity of DNA polymerase δ became defective *(pol3-01)* (Fig 6B, for every 10th base mismatched), suggesting that proofreading activity is responsible for incorporating mismatches as far as 40 nt. Consistent with previous observation, both evenly- and unevenly-mismatched donor templates showed significant ablation of mismatch incorporation in *pol3-01* background (Fig 6B and 6C). Even if the first 26 nt of the 3' invading end was identical to the donor, the assimilation of mismatches at the 27th base was dependent on Polδ proofreading.

Unexpectedly, we found that donor templates A and B, with a cluster of mismatches near the 3' invading end of the DSB, showed that the mismatch at the first base was still efficiently incorporated in *pol3-01*, even though all the other mismatches were generally not incorporated in this mutant (Fig 6C, donor template A and B). Previous study has shown that *pol3-01* is both almost completely exonuclease-deficient and strand displacement-proficient [37]. However, there may be some residual 3' to 5' proofreading exonuclease activity of DNA polymerase δ that could explain why we still see efficient incorporation of the first nucleotide mismatch in donor templates A and B (Fig 6C).

To confirm that assimilation of mismatches was independent of the Msh2/Mlh1-dependent mismatch repair system, we examined repair in the two reference strains, yRA280 and yRA321, as well as in donor template A (Fig 7A and 7B). In each case, cells lacking *MLH1* or *MSH2* still can extend and correct mismatches >30 bp from the 3' invading end of the DSB (Fig 7A and 7B). The mismatch assimilation of *msh2Δ* derivatives of remaining donor templates (templates B through F) with uneven distribution also did not proceed through Msh2-dependent mismatch repair (S3 Fig).

## Effect of the 3' flap endonuclease Rad1 on BIR with mismatched templates

The Rad1/Rad10 protein complex is required to remove 3' nonhomologous tails during strand annealing or strand invasion, to allow DNA polymerase to extend the paired 3' end [38–40]. Although in the design of the BIR substrates used here, there are no additional nonhomologous sequences at the HO cleavage site that would need to be removed, we asked if Rad1-Rad10 might still play a role in removing the 3' end when correction extends back as far as 20 to 40 bases. Deleting Rad1 did not affect the BIR efficiency or mismatch incorporation of mismatches on both evenly- and unevenly-mismatched donor templates (Fig 8A and 8B).

## Effect of Mph1 and Srs2 helicases on BIR with mismatched templates

In budding yeast, Mph1 is known to suppress BIR [41–43]. Previous *in vitro* studies suggested that Mph1 dissociates the invading strand of Rad51-generated D-loops or extended D-loops, possibly preventing inappropriate recombination or its resolution [44,45]. We confirmed that deleting Mph1 significantly increased the repair frequency by BIR (Fig 9A); however, there was no change in the pattern of incorporating mismatches into the BIR product (Fig 9B).

Srs2 is another DNA helicase that dislodges Rad51 from ssDNA to prevent any promiscuous strand invasions [46–48]. Srs2 is known to play a pro-recombinogenic role in DSB repair pathways via HR [49]. Our previous study showed that 98% of *srs2Δ* cells resulted massive cell

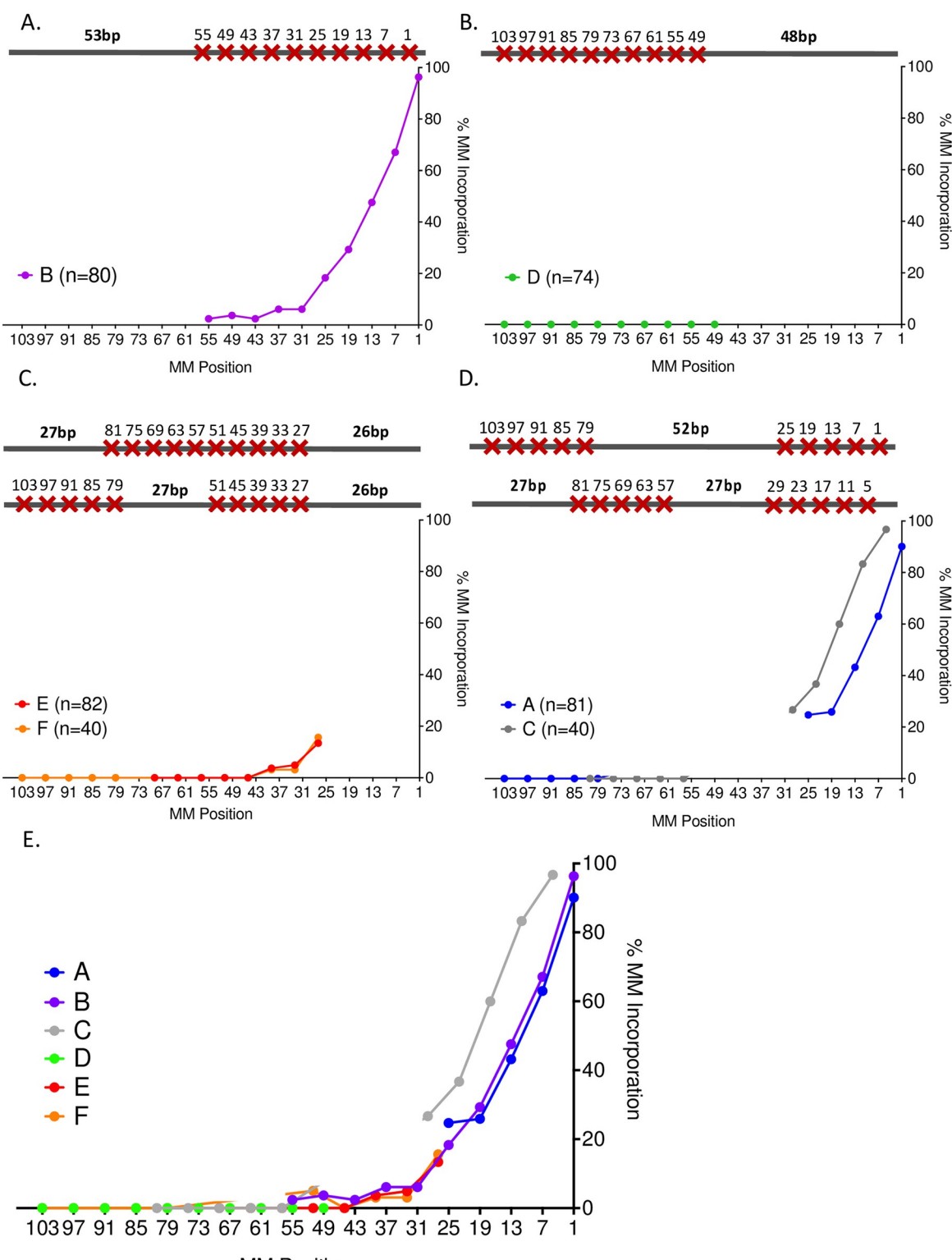

**Fig 5. Incorporation of mismatches into the BIR product.** A-D. Percent mismatch incorporation of individual donor templates containing 10 mismatches distributed unevenly throughout the 108-bp donor template. The number of sequenced recombinant clones for each template are indicated. E. Composite of % mismatch corrections for 6 different templates with different distributions of 10 mismatches.

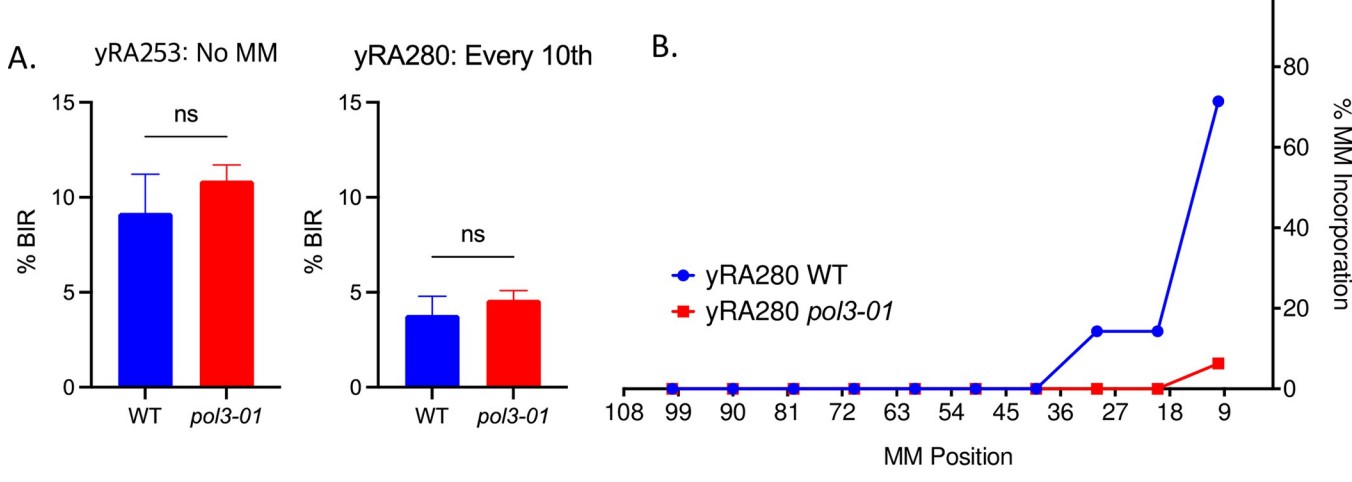

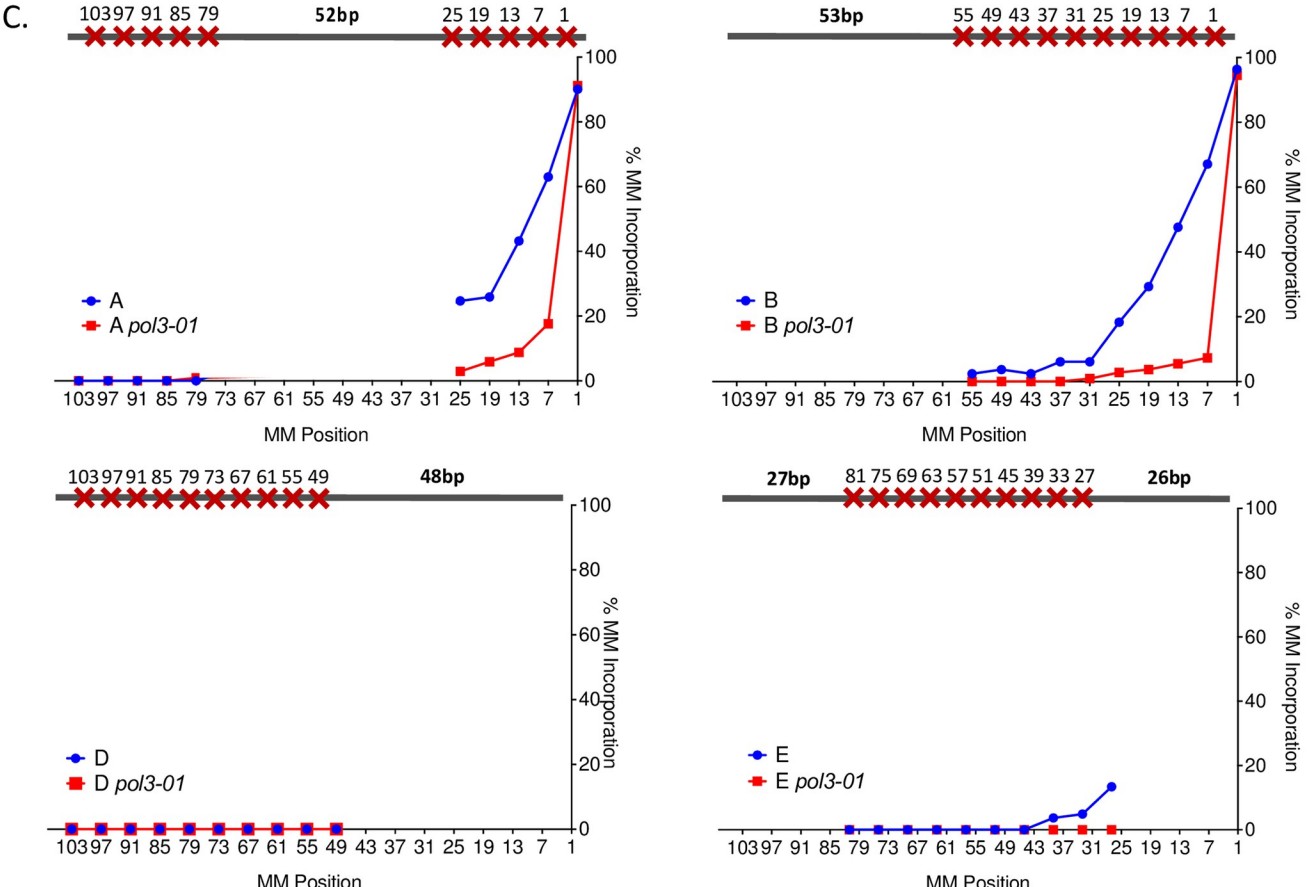

**Fig 6. 3' to 5' proofreading activity of DNA polymerase δ is responsible for incorporating most mismatches into the BIR product.** A. Effect of proofreading-defective DNA polymerase δ mutant (*pol3-01*) on BIR efficiency of donor templates with perfect homology (yRA253) and donor template mismatched every 10th bp (yRA280). Unpaired t-test with Welch's correction was used to determine the p-value. Error bars indicate standard deviation. A minimum of at least five measurements were performed. B. Percent mismatch assimilation of donor template mismatched every 10th bp (yRA280). A minimum of 40 samples were DNA sequenced. C. Percent mismatch assimilation of donor template with even mismatch distribution with proofreading-defective DNA Polymerase δ mutant (*pol3-01*). C. Effect of *pol3-01* on mismatch assimilation of donor templates with uneven mismatch distributions.

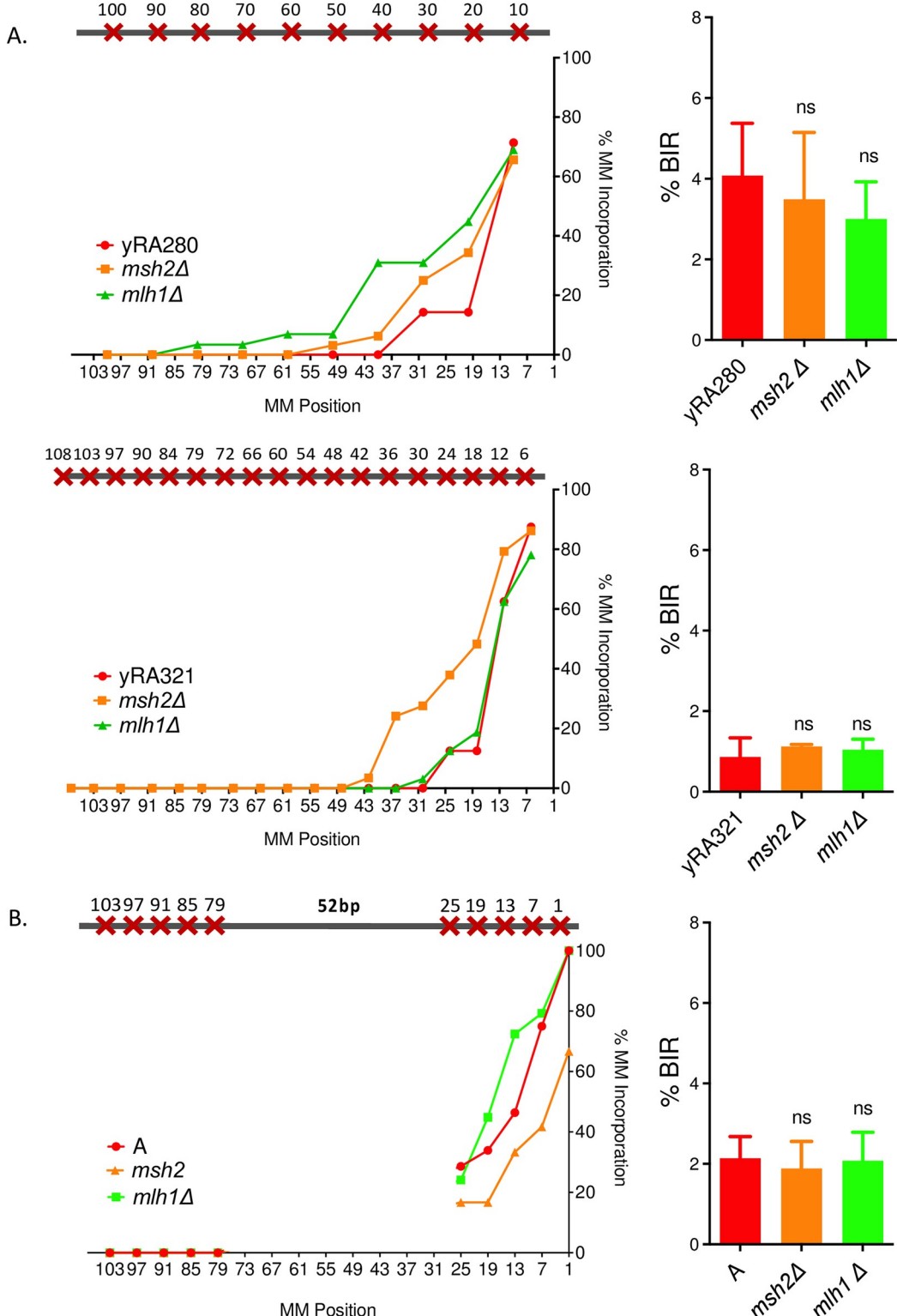

**Fig 7. The mismatch assimilation of the template sequence does not proceed through Msh2/Mlh1-dependent mismatch repair.** Effect of deleting mismatch repair genes *MSH2* or *MLH1* on mismatch incorporation pattern and BIR efficiency for strains yRA280 and yRA321 (A) or donor template A (B). Welch's t-test was used to determine the p-value. Error bars refer to standard error of the mean. A minimum of at least three measurements were performed. For all % mismatch incorporation data, a minimum of 40 samples were DNA sequenced.

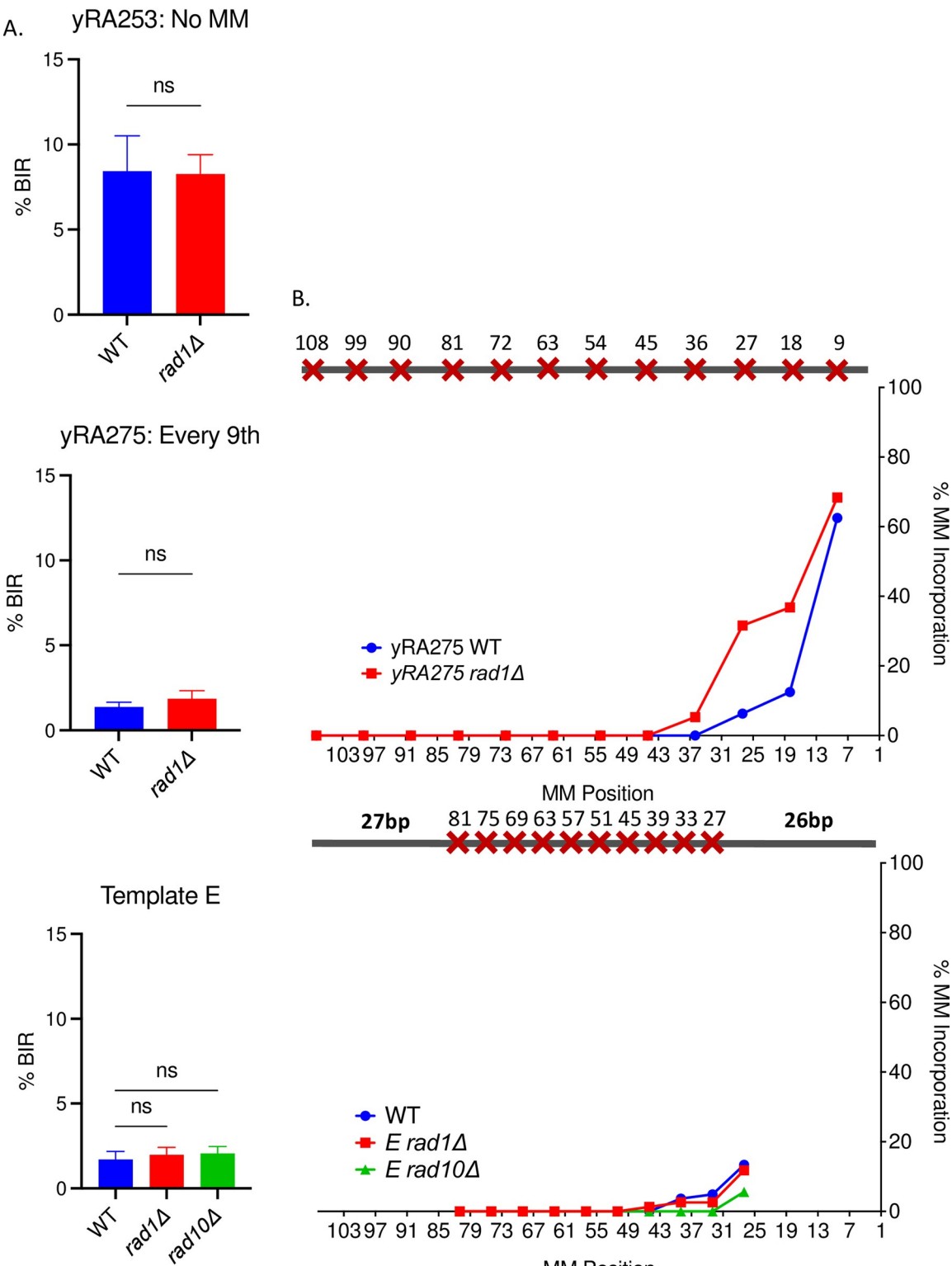

**Fig 8. The mismatch assimilation of donor sequences does not depend on Rad1-Rad10 flap endonuclease.** A. The effect of deleting the 3′-flap endonuclease *RAD1* or *RAD10* on BIR efficiency for yRA253 with mismatches every 9th bp, and donor template E. Welch's t-test was used to determine the p-value. Error bars refer to standard error of the mean. A minimum of at least five measurements were performed. B. The effect of deleting the 3′-flap endonuclease Rad1 or Rad10 on mismatch incorporation pattern for yRA253 (WT) with mismatches every 9th bp and donor template E. For all % mismatch incorporation data, a minimum of 40 samples were DNA sequenced.

death despite of successful completion of DSB repair [50]; lethality was attributed to the incomplete unloading of recombination factors, allowing Rad51 and RPA to bind persistently to the ssDNA even after the completion of the repair [51,52]. Here, we tested the role of Srs2 in Rad51-mediated BIR during mitotic recombination. Consistent with previous observations, deleting Srs2 significantly reduced the repair efficiency (S7 Fig). This observation agrees with a previous study that Srs2 is required for bubble migration during BIR and deleting Srs2 promotes formation of toxic joint molecules from uncontrolled Rad51 binding to the intact donor, interfering with BIR completion [53].

## Discussion

Although properties of budding yeast Rad51 have been well-studied *in vitro* [11,16,54,55], the ability of Rad51 to search and strand-invade donor sequences with mismatched sequences *in vivo* is not well understood. Here we have examined Rad51-dependent BIR where the 3' invading end of DSB and its donor templates share 108-bp homology, each carrying 10 mismatches, but arranged in several different ways with a spacing of every 6th bp. Donor templates that clustered their mismatches near the 3' end—where strand pairing must be accomplished before repair DNA synthesis can be initiated—were not less efficient in their repair compared to those with mismatches clustered at the 5' end or in other arrangements. Indeed, three templates with at least 26 nt of perfect homology at the 3' end were statistically significantly reduced in BIR. These results support *in vitro* studies that have suggested that Rad51-mediated pairing does not have to begin at the 3' end [9,11,12,56,57]; but it is not clear why substrates with a well-matched 3' end should be less efficient.

In the course of this work, we have resolved an apparent major difference in the measurement of tolerance of Rad51 for mismatched substrates *in vitro* versus *in vivo*. Previous *in vitro* studies had suggested that Rad51 was incapable of stably binding substrates in which there were fewer than 8 consecutive homologous base pairs [10,30,31], whereas *in vivo* we found significant levels of exchange when there are only 5 consecutive bases that can pair (every 6th base mismatched). A key difference between these findings is that Qi *et al.* [33] examined stable *in vitro* binding of a substrate that contained just a single tract of $\geq$ 8 base pairs of sequence homology, flanked by nonhomology, whereas the *in vivo* experiments presented in Anand *et al.* [17] and here, employed substrates that contained 18 tracts of 5 base pair homologies (i.e., 90 base pairs of total homology). Here, we show that Rad51, aided by Rad54, can indeed create stable D-loops *in vitro* with 90 or 108 nt ssDNA substrates in which every 6th base is mismatched (i.e., 75 or 90 base pairs of total homology, Fig 4A and 4D, respectively). Indeed, for templates containing only evenly-distributed mismatches, there is a very strong correlation (Pearson $R^2$ = 0.99) between *in vivo* and *in vitro* results for Rad51-mediated strand exchange (Fig 4C). Thus, a key finding of this work is that stable strand pairing can take place both *in vitro* and *in vivo* with substrates containing multiple tracts with fewer than 8 consecutive bases. We note that one caveat in comparing the results presented here to those from previous *in vitro* studies is that Rad54 was included in our gel-based D-loop assays but was not present in the prior DNA curtain studies [33]. The formation of D-loops in our gel-based assay is dependent on the presence of Rad54, hence Rad54 is required to detect stable reaction products in these assays. In contrast, the prior DNA curtain study was intended to detect both transient and stable reaction intermediates on the path towards D-loop formation [33].

Our data lend some support to the hypothesis that the success of strand pairing depends on the total number of base pairs that can be formed, or–more precisely–to the total energy of base pairing that is achieved. Thus, among the set of 6 donors with different clustering of mismatches, those with a higher Tm tended to have higher BIR efficiencies. The relationship

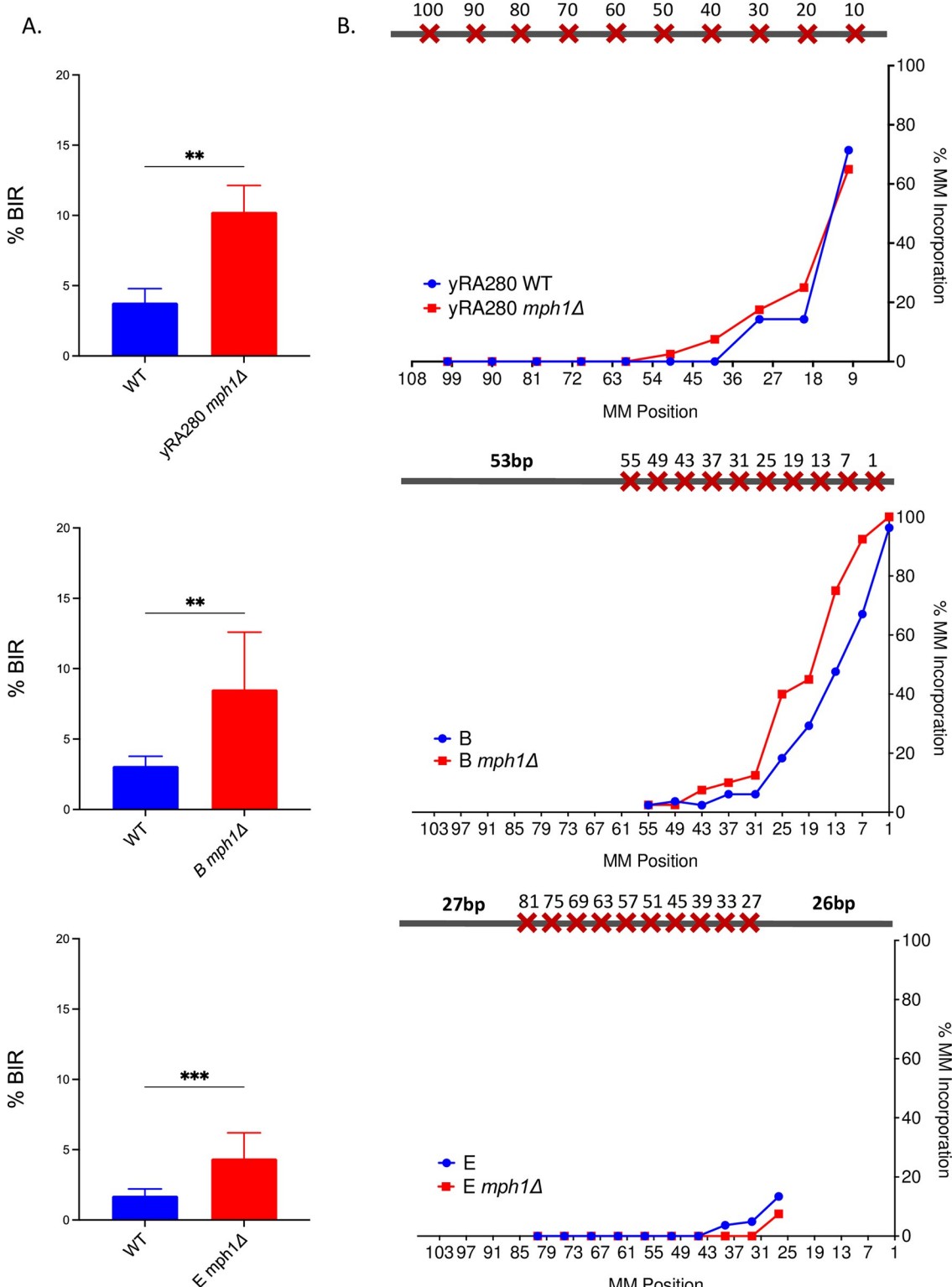

**Fig 9. DNA helicase Mph1 suppresses BIR efficiency.** A. The effect of deleting the DNA helicase Mph1 on BIR efficiency for yRA280 and donor templates B and E. Welch's t-test was used to determine the p-value. Error bars refer to standard error of the mean. A minimum of at least five measurements were performed. B. The effect of deleting Mph1 on mismatch incorporation pattern for yRA280, donor templates B and E. For all % mismatch incorporation data, a minimum of 40 samples were DNA sequenced.

between template Tm is the same for the six unevenly-distributed donors compared to the evenly-distributed controls, but the set with clustered mismatches is repaired less efficiently. This downward shift is a feature of the *in vivo* process, as the *In vitro* analysis of these same templates shows a similar relationship between pairing efficiency and Tm but does not show a reduction of D-loop formation by unevenly-mismatched substrates relative to the evenly-matched controls (Fig 3C). A comparison of the data gathered *in vitro* and *in vivo* for the 6 unevenly-distributed substrates revealed that the two data sets are statistically significantly different. These results suggest that there are impediments to *in vivo* recombination compared to *in vitro* D-loop formation. It is, however, not yet evident why there is such a large range of outcomes for these different arrangements. It is possible that there is some sequence-specific secondary structure of the single-strand oligonucleotide but we have not been able to deduce such a motif. It is also unclear what enforces the reduced success of these templates *in vivo*, but it is independent of the Msh2 mismatch repair system that might promote heteroduplex rejection. It is possible that there is some specific mismatch or set of mismatches that either create some novel secondary structure or are recognized by a specific DNA binding protein.

### Role of DNA polymerase δ in the incorporation of donor sequences into the BIR product

The second finding we draw from these experiments concerns the action of DNA polymerase δ and its 3' to 5' proofreading activity. The pattern of mismatch incorporation into the BIR product was the same at interior positions whether or not they were preceded by mismatches nearer the 3' end; that is, DNA Pol δ still "backs up" with 3'→ 5' exonuclease activity to incorporate mismatches even when the first 26 positions are completely homologous. Thus, proofreading at these interior positions is not dependent on any prior mismatch cues and suggests that Pol δ intrinsically backs up on DNA templates even when they are fully homologous. This activity is far more extensive than its action in removing a single mismatched base during DNA replication and is more comparable to the ability of Pol δ to excise a 3'-ended nonhomologous tail at the end of a DSB being repaired by gene conversion [54]; however, the pattern of mismatch assimilation is not compatible with an intermediate in which an entire unpaired mismatched end would be excised as a 3' flap, such that the incorporation of mismatches would be the same at each position. Deletion of *MSH2* or *MLH1* had no significant effect on the repair outcome, nor did removal of the 3' flap endonuclease, Rad1-Rad10.

It is not necessary that Pol δ removes 27 bases in a single encounter. It is possible that Pol δ removes only one or a few bases but does so in repeated encounters with the 3' end before it initiates new DNA synthesis. Such a reiterative process could explain the surprising finding that *pol3-01* does not impair the removal of the first base at the 3' end of the invading strand while blocking the removal of mismatches further along. Although *pol3-01* appears *in vitro* to be unable to excise a single mismatched base from a paired primer before new DNA synthesis is initiated [37], we entertain the idea that a small amount of residual 3' to 5' excision activity might remain and that multiple cycles of binding of Pol δ to the 3' end might result in the removal of the most terminal mismatch. Alternatively, there may be another exo- or endonuclease activity that can accomplish this end-removal.

## Methods

### Yeast strains

yRA strains used in these experiments have the haploid S288c background (*ho mat*Δ::hisG *hml*Δ::hisG *hmr*Δ::*ADE3 ura3*Δ-851 *trp1*Δ-63 *leu2*Δ::KAN *ade3::GAL10::HO*), in which the

site-specific HO endonuclease is under control of a galactose-inducible promoter [36]. Strain yRA111 contains the recipient sequences composed of the 5' sequences of *URA3* gene (UR), an artificially inserted split-intron with the splice donor site (5' SD) and the HO recognition site (HOcs), located at the *CAN1* locus in a non-essential terminal region of chromosome V [17]. Divergent donor sequences containing uneven distribution of mismatches were designed and ordered as synthetic gBlock gene fragments from IDT and assembled into pRS314-based (*CEN4*, *TRP1*) plasmid bRA29 containing the 3' splice-acceptor (3' SA) of the intron, the 3' sequence from the *URA3* gene (A3) and the *TRP1* marker using *in vivo* recombination and plasmid rescue from a yeast host strain as described in Anand et al. [17]. The donor cassettes were PCR-amplified from plasmid bRA29 and integrated at the *FAU1* locus, about 30 kb from the right end of on chromosome V. Further information for strains, recipients (donors) and plasmids are found in S1, S2, S3 and S4 Tables.

## BIR assay

Selected strains were grown on YEPD (1% yeast extract, 2% bactopeptone, 2% dextrose) + ClonNAT (100uL/mL final concentration) at 30˚C. Individual colonies from strains were picked and serially diluted 1000-fold in double-distilled H$_2$O. Serial dilutions were plated on YEPD to obtain the total number of cells and on YEP-galactose (YEP-Gal) to measure the recombination-dependent survivors after inducing HO endonuclease expression [36]. Cells were incubated at 30˚C for 2–3 days. Cells on YEP-Gal were then replica plated to plates lacking uracil, to count colonies that survived the break via BIR, and to Nourseothricin (NAT) plates, to count colonies which survived the break via nonhomologous end-joining (NHEJ) that alters the HO cleavage site but retains the distal part of the left arm of chromosome V [3]. Ura⁻ NAT⁺ colonies, arising by nonhomologous end-joining, arose at a frequency of approximately 0.5% of the total number of colonies and are not reported for each assay.

## DNA sequence analyses of the break repair junctions

Representative Ura⁺ colonies that had repaired by BIR were confirmed by PCR, using primers amplifying the region at the start and end of the *URA3* gene using primers DG31 (GGAACGTGCTGCTACTCATC) and DG32 (TTGCTGGCCGCATCTTCTCA). PCR products were initially checked by gel electrophoresis and sent to GENEWIZ for Sanger sequencing. Individual PCR sequences were aligned with corresponding 108-bp donor templates and analyzed by DNA analyses software Serial Cloner 2.6.1 and Geneious Prime software.

## Statistical analysis

GraphPad Prism 9 software was used to calculate statistical significance of data, based on Dunnett's comparison of multiple samples versus a single control. Thermal stability, as reflected in melting temperature Tm (˚C). of the base-pairing between the DSB 3' and a complementary single strand of the donor template was determined using the method of Markham and Zuker (http://www.unafold.org/Dinamelt/applications/two-state-melting-hybridization.php) [37]. Statistical differences between *in vivo* and *in vitro* results for the six arrangements of mismatched bases (A though F) were determined by a Mann-Whitney test arbitrarily pairing each of 4 independent measurements of the *in vivo* set with the results of the *in vitro* set. By using Geneious Prime's DNA secondary structure fold viewer (https://assets.geneious.com/manual/2021.1/static/GeneiousManualse36.html), we monitored possible secondary structures of the donor templates. ΔG of all donor templates were calculated to determine the energy required to break the secondary structure.

## D-loop assays

D-loop assays using supercoiled plasmid and purified proteins were performed as previously described [58]. Recombinant yeast Rad51, Rad54, and RPA were expressed and purified as before [59]. 5' fluorescently-labeled 90-nt or 108-nt ssDNA probes were ordered from IDT and gel-purified prior to use. Assays were started by first incubating 10 nM ssDNA with stoichiometric amount of Rad51 at 30°C for 15 min in reaction buffer containing 30 mM Tris-Acetate pH 7.5, 20 mM Mg-Acetate, 50 mM KCl, 0.1 mg/mL BSA, 1 mM DTT, and 5 mM ATP. The pre-formed Rad51 filaments were then mixed with 9.25 nM supercoiled (CURMID-Curtains Plasmid) plasmid [59] or pUC19_108 for 90mer or 108mer reactions, respectively), 750 nM RPA, and 95 nM Rad54 to initiate strand-exchange. Reactions were incubated at 30°C, stopped after 5 minutes by adding equal volumes of buffer containing 20% glycerol, 1% SDS, and 25 mM EDTA, and deproteinized with 1/10 volume of Proteinase K (NEB P8107S) at 37°C for 30 min. Reaction products were resolved on 0.9% TAE-agarose gel and scanned with a Typhoon FLA 9000 (GE Healthcare).

## Supporting information

**S1 Fig. Rad51-mediated Break-Induced Replication.** A. Mechanism of Rad51-dependent BIR B. Mismatch incorporation of heteroduplex DNA formation during BIR. Once a DSB is created, a broken end of DSB will be resected by 5'→3' exonuclease to generate 3' single-stranded DNA (ssDNA) which interacts with Rad51 and other recombination proteins to carry out homology search and strand invasion. The resected end of recipient sequence (indicated in red) will synapse with the donor (indicated in blue). Mismatches in the heteroduplex region during strand invasion are apparently not corrected by the Msh2/Mlh1 mismatch repair complex; rather DNA polymerase δ is recruited to the 3' end and performs its proofreading 3'→5' exonuclease activity prior to initiating new DNA synthesis from the 3' invading end. DNA Polymerase δ "can apparently "back up" into the heteroduplex region as far as 40–50 nt and resynthesizes the region, copying the donor template sequences into the recovered BIR product.
(TIFF)

**S2 Fig. The effect of *MSH2* mutants on the repair efficiency.** Wild type and *msh2Δ* derivatives of each donor template were measured as described in Fig 1. Statistical significance of the differences for each donor/*msh2Δ* pair was determined using an unpaired t-test with Welch's correction. Error bars refer to standard deviation. Each measurement is based on a minimum of three experiments.
(TIFF)

**S3 Fig. The mismatch assimilation of *msh2Δ* derivatives of donor templates with uneven mismatch distribution also does not proceed through Msh2-dependent mismatch repair.** The effect of deleting mismatch repair gene *MSH2* on mismatch incorporation pattern and BIR efficiency for donor templates with uneven mismatch distribution. For all % mismatch incorporation data, a minimum of 40 samples were DNA sequenced.
(TIFF)

**S4 Fig. Differences in BIR efficiency for wild-type donor templates in relation to each other.** Donor template were compared among them to assess the statistical significance of different arrangements of clustered mismatches for BIR efficiency. Significance determined using a Tukey's multiple comparison tests (GraphPad Prism 9). Error bars refer to standard deviation. ** $p < 0.001$, ns = not significant. % BIR graph only indicated statistical significance and

not included ns.
(TIFF)

**S5 Fig. Differences in BIR efficiency for *msh2Δ* derivatives of donor templates in relation to each other.** Each *msh2Δ* derivative of donor templates was compared between them to assess the statistical significance of different arrangements of clustered mismatches for BIR efficiency. Significance determined using a Tukey's multiple comparison tests (GraphPad Prism 9). Error bars refer to standard deviation. ** $p<0.001$, ns = not significant. All data on % BIR graph for *msh2Δ* derivatives of donor templates were not significant when compared between them.
(TIFF)

**S6 Fig. BIR efficiency for *pol3-01* mutants of donor templates with uneven mismatch distribution.** Percent BIR efficiency of donor templates A, B, D, and E with proofreading-defective DNA Polymerase δ mutant (*pol3-01*). Unpaired t-test with Welch's correction was used to determine the p-value. Error bars indicate standard deviation.
(TIFF)

**S7 Fig. BIR efficiency for *srs2Δ* mutants.** Percent BIR efficiency of yRA280 and donor templates B and E with *srs2Δ*. Unpaired t-test with Welch's correction was used to determine the p-value. Error bars indicate standard deviation.
(TIFF)

**S1 Data. All of the data concerning the efficiency of BIR and the percent of mismatch incorporation in various mutant backgrounds are presented in the dataset.**
(XLSX)

**S1 Table. Yeast strains used in this study.**
(DOCX)

**S2 Table. Recipient and donor sequences.**
(DOCX)

**S3 Table. Primers used in this study.**
(DOCX)

**S4 Table. Plasmids used in this study.**
(DOCX)

**S5 Table. Sequences of 90-nt ssDNA containing evenly-distributed mismatches.**
(DOCX)

## Acknowledgments

Special thanks to Paul Miller, Michael Lichten, Stephen Levene, Myron Goodman and Chiho Mak for discussions about this research.

## Author Contributions

**Conceptualization:** Jihyun Choi, Kevin Li, James E. Haber.

**Data curation:** Jihyun Choi, Muwen Kong.

**Formal analysis:** Muwen Kong, Eric C. Greene, James E. Haber.

**Funding acquisition:** Eric C. Greene, James E. Haber.

**Investigation:** Jihyun Choi, Yiting Cao.

**Methodology:** Jihyun Choi, Muwen Kong, Danielle N. Gallagher.

**Project administration:** Jihyun Choi, James E. Haber.

**Resources:** Jihyun Choi, James E. Haber.

**Software:** Jihyun Choi.

**Supervision:** Eric C. Greene, James E. Haber.

**Validation:** Jihyun Choi, Eric C. Greene.

**Visualization:** Jihyun Choi.

**Writing – original draft:** Jihyun Choi, James E. Haber.

**Writing – review & editing:** Jihyun Choi, Muwen Kong, Danielle N. Gallagher, Kevin Li, Gabriel Bronk, Eric C. Greene, James E. Haber.

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
