## [Decision Letter · Decision Letter 0]

28 Feb 2022

Dear Dr Haber,

Thank you very much for submitting your Research Article entitled 'Repair of Mismatched Templates during Rad51-dependent Break-Induced Replication' to PLOS Genetics.

The manuscript was fully evaluated at the editorial level and by three independent peer reviewers who are experts in the field. The reviewers appreciated the attention to an important problem, but raised some substantial concerns about the current manuscript. Based on the reviews, we will not be able to accept this version of the manuscript, but we would be willing to review a much-revised version. We cannot, of course, promise publication at that time.

Reviewer #1 is the most critical but all three reviewers point to lack of precision in the writing, an over-use of generalizations and a failure to consider alternative models. 

If you decide to revise the manuscript for further consideration at PLOS Genetics, please aim to resubmit within the next 60 days, unless it will take extra time to address the concerns of the reviewers, in which case we would appreciate an expected resubmission date by email to plosgenetics@plos.org.

[LINK]

We are sorry that we cannot be more positive about your manuscript at this stage. Please do not hesitate to contact us if you have any concerns or questions.

Yours sincerely,

Gregory P. Copenhaver

Editor-in-Chief

PLOS Genetics

Gregory Barsh

Editor-in-Chief

PLOS Genetics

Reviewer's Responses to Questions

**Comments to the Authors:**

Reviewer #1: This manuscript will be of interest for researchers in the area of genome stability and mechanisms of homology-directed DNA repair. The study follows an earlier high-profile paper (Anand et al. 2017 Nature), where the Haber laboratory identified an astounding mismatch tolerance of Rad51-mediated break-induced replication. Here, the authors conduct careful analysis of additional mismatched constructs that test hypotheses that length and position of uninterrupted homology affect mismatch tolerance. The analyses are conducted in wild type and mismatch repair-deficient strains disabling mismatch recognition (msh2) and mismatch processing (mlh1). Sequence analysis of individual recombinants is used to assess the correction of the mispairs formed during strand invasion. The experiments are well designed and executed, but the results limit potential conclusions, and the work does not generate novel insights. A number of the conclusions appear overly generalized and do not consider the unusual properties of the system used here. In particular, the main conclusion that the exonuclease activity of DNA polymerase delta can chew back for 26 nt without mismatch cues is overstated and an alternative model of reiterative invasion and chew-back should be considered and experimentally tested.

Major comments:

1) In the introduction (line 77 ff), the authors discuss the properties of their assay system and correctly point out that it is above the minimal homology requirement for recombination in yeast. However, the authors should also discuss the differences to system with longer homology and infinite homology, as it would be the case for allelic HR between sisters/homologs. In other words, the authors should acknowledge the limitations of the system sharing only 108 bp homology. Moreover, the authors assume that the D-loops in their study are uniform in structure and length, all invading with their 3’-end, although it is currently not possible to analyze these D-loop properties in vivo. Furthermore, the authors neglect to discuss discrepancies between their findings and other published studies. Guo et al. 2017 Mol Cell conducted a similar study with dramatically different conclusions as to the role of Pol delta in 3’ mismatch correction. A discussion of this study and other similar studies should be included to contextualize the findings presented in Choi et al. in the broader HR field.

2) Figure 2 and lines 82, 83, and throughout: “approximately 14% of cells successfully repaired a DSB when the homologous region is 108 bp”, this value differs from what is shown in Figure 2B, which shows ~8-9% of cells repairing by BIR. The text should be corrected to reflect the data reported in the figure.

3) Figure 2 and lines 87-90, 129, 130: “Repair efficiency was significantly reduced but still significant when every 6th base was mismatched”; the authors should reference Figure 2B. Furthermore, it is debatable whether a repair event observed in only 1-2% of the population is “significant.” The authors should give the actual percent of cells completing repair. Considering repair is almost negligible in this system when every 6th bp is mismatched, it is misleading to claim that their in vivo results differ from results obtained by in vitro studies.

4) Figure 2 and Line 101: “but exhibit significant differences that depend on the precise location of the mismatches”; not all the differences are significant, and the authors fail to interpret their findings in a way that makes a convincing argument that the position of the mismatches influences the outcome in a predictable or logical manner. The results simply do not lend themselves to making significant conclusions about position of uninterrupted homology.

5) Figure 3 and lines 157-161: msh2 clearly effects the efficiency of BIR in a substrate-dependent manner according to Figures 3 and S2. The perfectly matched donor (yRA253) shows decreased BIR in the msh2 background. However, the donor with every 6th bp mismatched (yRA321) shows somewhat increased BIR in the msh2 background. Furthermore, the substrates with 3’ mismatches (‘B’), with mismatches in the middle of the donor (‘E’), and with mismatches at the 5’ end and in the middle of the donor (‘F’) also have slightly increased BIR in the msh2 background. In addition, the slope of the line in Figure 3B flattens relative to Figure 3A. This indicates that while msh2 does not fully suppress the differences between the evenly-spaced mismatches and the unevenly-spaces mismatches, at least some of the differences are partially suppressed. The authors need to revise these lines to better fit their findings.

6) Figures 4, 5 and Lines 108-111, 166-168, 232, 233: The authors could have made this point more convincingly by using the pol3-5DV mutant, as in Anand et al. 2017 and Guo et al. 2017 Mol Cell. Moreover, though the authors’ results on Pol delta’s proofreading activity differ significantly from the findings in Guo et al., the authors do not cite or discuss this paper in the text.

The effects of msh2 and mhl1 demonstrated in Figure 5 are neither acknowledged nor discussed. Lines 189, 190 should be repharsed and expanded.

7) The major conclusion of the manuscript is that DNA polymerase delta exonuclease can chew back 26 nt with mismatch clues. This conclusion is not well supported by the analysis. Can the authors exclude alternative mechanisms or models? Is it possible that 3’-flappases like Rad1-Rad10 or Mus810-Mms4 play a role? Is it possible that he results are a reflection of reiterative cycles of invasion, chew-back and D-loop dissolution? This would be consistent with the expected instability of the short D-loop, the evidence for reiterative invasion cycles and the known proofreading window for Pol delta exo, which is significantly smaller than 26 nt. Moreover, this would be more consistent with the previous analysis by Guo et al. This possibility should be explored using mutants that limit D-loop dissolution (srs2, mph1, sgs1), and the results may drastically change the major conclusion.

Additional comments:

8) Figure 1, Table 1: Is pairing possible between the UR of the assay system and ura-3D851, also present in these strains? This could significantly influence the interpretation of the results. Even if there is no homology, the authors may want to state this clearly.

9) Lines 54-57: NHEJ and alt-EJ are generally considered distinct pathways for DNA damage repair/tolerance.

10) Lines 59, 60: Single-strand annealing (SSA) is a homology-directed repair pathway but not typically considered an HR sub-pathway. The discussion of DNA damage repair/tolerance pathways and HR sub-pathways could be re-written and expanded to include more relevant details.

11) Lines 64-72: The description of BIR includes certain very specific details while glossing over other key steps (e.g. Rad51 mediators and accessory factors, Pol32 requirement). The description of BIR could be revised to include only the steps that are relevant to understanding the findings presented in this paper, and should be congruent with the diagram in Figure S1 (e.g. Rad52 is shown in the Figure but not discussed in the text).

12) Lines 78, 79: Both ends of the break do not necessarily interact with the donor during SDSA,

13) Line 111: “There was little effect when Msh2/Mlh1-dependent mismatch repair was ablated”; the results presented in Figure 3 indicate that there is an effect of msh2.

14) Lines 130, 131: “BIR…still occurred about 9% in yRA321”, this makes it sound as though 9% of the population completes repair by BIR with the yRA321 substrate, when in fact BIR is reduced to 9% of the level observed in yRA280. 9% repair by BIR would be a significant level of repair, but 1-2% repair is not necessarily significant. The authors need to clarify their results so as not to mislead the reader.

15) Lines 168,169: The authors should cite Guo et al. 2017 Mol Cell.

16) Lines 225-228: “Indeed, deleting Msh2 led to an overall reduction in repair efficiency for both the evenly- and unevenly-spaced templates,” this statement is not quite correct based on their own results in Figures 3B and S2. BIR efficiency increased in the msh2 background for some of the mismatched donors, whether the donors had evenly spaced mismatches (yRA321) or unevenly spaced mismatches (‘A’, ‘E’, and ‘F’).

17) Figures 2, 3: Why is the color indicating construct ‘A’ different between figures? The color representing ‘A’ should be consistent.

18) Table 1: Generally, ‘::’ denotes that a gene has been deleted and replaced with another marker. Thus, while I infer that “can1DEL::UR intron_SD::HOcs::NAT” is the site of the DSB in their assay system, it might be more appropriate to write this as “can1::UR intron-SD-HOcs-NAT”. This is especially confusing considering another maker present in these strains is HOcs::hisG. In this case, I infer that the HOcs at its native locus has been deleted, but given the annotation of the assay system HOcs, it is unclear whether this is the case.

Reviewer #2: Please see the attachment

Reviewer #3: In the reviewed manuscript entitled “Repair of Mismatched Templates during Rad51-dependent Break-Induced Replication” the authors presented the results of their studies on the efficiency of break-induced replication (BIR), the homologous recombination subpathway. In their study they explored the influence of mismatches distribution in a donor template on the final repair product. By examining different templates containing the same number of mismatches but distributed unevenly, they were able to show the polar effect of insertion of the donor sequence into the final repair product. The mismatches located at the 3’ end of the invading strand were commonly replaced by the template sequence, but this tendency was gradually limited, and about 40 bp from the 3’ end, incorporation of the template sequence was rare. According to the authors, this effect was linked to the 3’-5’ exonucleolytic (proofreading) activity of polymerase δ, synthesizing DNA during repair; however, it had nothing to do with the proofreading activity per se, since 3’-5’ exonucleolytic activity was observed independently of mismatches presence or absence in the sequence. Authors call this activity “chewing back” the 3’ end of invading strand and claim that this step precedes the initiation of DNA synthesis using the donor template. Thus, the additional step during the initiation of BIR was recognized. The finding that donor strand undergoes resection is important; however, an indication of which enzyme is responsible for its processing is still open. The authors did not prove that, so they should moderate their statement and include alternative scenarios in the text or make a similar set of experiments as they described, using a set of strains with a polymerase variant impaired in proofreading activity. In the previous works, the researchers from the same laboratory showed that it is possibly not Exo1 activity; however, yeast Saccharomyces cerevisiae possesses broader resources of 3’-5’ exonucleases that should be considered.

Interestingly, one of the findings is that the Msh2-dependent mismatch repair system seems not to contribute to BIR even when mismatches are present in the duplex created by the acceptor and donor strand. Presented data contribute to our understanding of the recombination process. However, the results of this study also underlined the differences between results obtained in the in vitro ad in vivo studies and showed that the model used to reveal the mechanism of recombination could influence the results by the length and sequence of the homologous region used in the attempt.

The data obtained by the authors brought us closer to solving the complicated puzzle of the homologous recombination process, whose final effect is enzyme- and template-dependent.

**Have all data underlying the figures and results presented in the manuscript been provided?**

Reviewer #1: Yes

Reviewer #2: Yes

Reviewer #3: Yes

PLOS authors have the option to publish the peer review history of their article (what does this mean?). If published, this will include your full peer review and any attached files.

Reviewer #1: No

Reviewer #2: No

Reviewer #3: No

---

## [Decision Letter · Decision Letter 1]

27 Jul 2022

Dear Jim,

Thank you very much for submitting your Research Article entitled 'Repair of Mismatched Templates during Rad51-dependent Break-Induced Replication' to PLOS Genetics.

The manuscript was fully evaluated at the editorial level and all 3 of the original independent peer reviewers. Reviewers #1 and #3 are now satisfied, but Reviewer #2 has a list of minor concerns that should be addressed.  Once I receive your revised manuscript I will be able to render and editorial decision without further external review.

We therefore ask you to modify the manuscript according to the review recommendations. Your revisions should address the specific points made by each reviewer.

[LINK]

Yours sincerely,

Gregory P. Copenhaver

Editor-in-Chief

PLOS Genetics

Gregory Barsh

Editor-in-Chief

PLOS Genetics

Reviewer's Responses to Questions

**Comments to the Authors:**

Reviewer #1: Manuscript Title: Repair of Mismatched Templates during Rad51-dependent Break-Induced Replication revised

Authors:

Jihyun Choi; …… James E. Haber

The revised manuscript is significantly improved and addresses the concerns of the review. Moreover, the authors included data from Eric Greene’s laboratory to largely resolve the conflict between in vivo and in vitro data regarding mismatch tolerance.

Reviewer #2: Presented revised manuscript contains two sets of new results:

• analysis of pol3-01 mutant demonstrating that it is indeed the exonuclease activity of DNA polymerase delta that is responsible for extensive substrate degradation even when no mismatches are present

• in vitro D-loop assays with evenly and unevenly mismatched templates that complement presented data on the effect of mismatches on BIR in vivo

These are very valuable inclusions that greatly enhance the value of this manuscript.

However, some general conclusions are made based only on the subset of presented data and discussion of unexpected findings is very brief.

Comment 1. Generalized conclusions, understatements based on a subset of presented data. These can be very misleading to the reader.

line 23 These different arrangements of uneven mismatch distributions were in general less efficient in recombination than templates with evenly distributed mismatches.

2 (B, C) out of 6 were not less efficient, they were as efficient as evenly-spaced control.

The Authors acknowledge that

line 29 mismatch position-specific effects are also important

but continue to generalize without making reservations about B and C (also lines 235, 345, maybe others), and such imprecise statement will remain as take home message from this paper.

line 25 A donor with all 10 mismatches clustered at the 3’ invading end of the DSB was not impaired compared to arrangements where mismatches were clustered at the 5’ end.

Understatement, donor B was not impaired not only when compared to donor D (which was one of 4 impaired as compared to the evenly-spaced control), but also when compared to the evenly-spaced control itself.

line 214 (also line 347) But unlike the results in vivo, there was no general reduction in product formation compared to the evenly-distributed controls in the in vitro assays.

Understatement, in fact BIR efficiency was increased for 4 out of 6 substrates, same for C and lower for E. No statistical significance shown for in vitro data (such as seen for in vivo BIR in Fig2C).

line 228 In general, the efficiency of BIR was lower in the absence of Msh2 and line 235 We conclude that there is some other factor beyond the mismatch repair machinery that causes the unevenly distributed mismatches to be less successful in BIR than the evenly-distributed controls

Both relate to the Authors previous response to Rev 1 comment 16.

I agree WT and delmsh2 sets may be different, but if so why the conclusion is that the difference is mediated by other factors than Msh2?

I am not convinced it is appropriate to describe data presented in Fig. S2 (as well as other data in this manuscript, as mentioned above) in terms of “data as a whole” or „in general”.

This obscures a fact that some donors with unevenly-spaced mismatches do appear to behave in a different, difficult to explain but, in my opinion, potentially very interesting way.

 

Comment 2.

line 282 ...cells lacking MLH1 or MSH2 still can extend and correct mismatches >30 bp from the 3’ invading end of the DSB...

Is the drop to 60% for incorporation of first mismatch for template A in delmsh2 meaningful?

Comment 3.

line 293 Deleting Rad1 did not affect the BIR efficiency or mismatch incorporation of mismatches on both evenly- and unevenly-mismatched donor templates.

Generalisation based on one unevenly-mismatched donor. Why yRA275 and not yRA280 was used as evenly spaced control? Why E was chosen? It would be interesting to see if first mismatch is affected in donor A (as examined for MSH2) or B (as examined for MPH1).

Same representative subset of substrates should be examined for all mutants tested and their choice explained.

Comment 4.

line 293 This observation agrees with a previous study that Srs2 is required for bubble migration during BIR and deleting Srs2 promotes formation of toxic joint molecules from uncontrolled Rad51 binding to the intact donor, interfering with BIR completion.

Presented data do not relate to the mechanism of Srs2 activity. They show that in delsrs2 the frequency of BIR is too low to assess the effect of this helicase on BIR with mismatched templates.

Comment 5.

line 322 These results support in vitro studies that have suggested that Rad51-mediated pairing does not have to begin at the 3’ end (9, 11, 12, 55, 56), but it is not clear why substrates with a well matched 3’ end should be less efficient.

Perhaps it is not that well matched 3' end is less efficient but that less matched further regions (that arise if mismatches are clustered away from 3'end) are important in vivo. This would be an important phenomenon that failed to be captured by in vitro assays.

Comment 6.

line 326 Previous in vitro studies had suggested that Rad51 was incapable of stably binding substrates in which there were fewer than 8 consecutive homologous base pairs....

and line 334 Here, we show that Rad51, aided by Rad54, can indeed create stable D-loops in vitro with 90 or 108 nt ssDNA substrates in which every 6th base is mismatched.....

Was Rad54 included in earlier cited in vitro studies? If not this would be another difference between previous and current work that should be discussed.

Comment 7.

line 178 The differences among these templates cannot be attributed to a difference in thermal stability of base-pairing in the 108-bp region as measured by the calculated melting temperature (Tm) between complementary 108-nt DNA strands.... Interestingly, the slopes of the linear regression lines were nearly identical for the evenly- distributed controls .... as for the 6 unevenly spaced cases .... but the unevenly-distributed series lie below the controls.

Description of data presented in Fig.3 is very brief and it is not apparent how the Authors reach and what they mean (with respect to mismatched templates that are examined in this work) by the general conclusion presented in the Discussion :

line 341 Our data lend some support to the hypothesis that the success of strand pairing depends on the total number of base pairs that can be formed, or – more precisely – to the total energy of base pairing that is achieved.

Comment 8.

line 341 It is also unclear what enforces the reduced success of these templates in vivo...

This is a very interesting finding that perhaps warrants further discussion. Other helicases, chromatin factors that could play a role and be examined in the future?

Comment 9.

Discussion of another very interesting result is also brief.

line 380 ... we entertain the idea that a small amount of residual 3’ to 5’ excision activity might remain...

On what basis?

line 383 ....there may be another activity that can accomplish this end-removal.

Such as what?

Comment 10.

line 132 ....but, unexpectedly, donor templates with majority of mismatches towards the 5’ end were statistically significantly lower than the evenly-distributed control.

Unclear, needs rephrasing.

Reviewer #3: In the revised version of the article, the Authors addressed all my comments and suggestions, therefore the manuscript is now suitable for publication.

**Have all data underlying the figures and results presented in the manuscript been provided?**

Reviewer #1: Yes

Reviewer #2: Yes

Reviewer #3: Yes

PLOS authors have the option to publish the peer review history of their article (what does this mean?). If published, this will include your full peer review and any attached files.

Reviewer #1: No

Reviewer #2: No

Reviewer #3: **Yes: **Adrianna Skoneczna

---

## [Editor Report · Decision Letter 2]

10 Aug 2022

Dear Dr Haber,

We are pleased to inform you that your manuscript entitled "Repair of Mismatched Templates during Rad51-dependent Break-Induced Replication" has been editorially accepted for publication in PLOS Genetics. Congratulations!

Yours sincerely,

Gregory P. Copenhaver

Editor-in-Chief

PLOS Genetics

Gregory Barsh

Editor-in-Chief

PLOS Genetics

Comments from the reviewers (if applicable):

**Data Deposition**

http://datadryad.org/submit?journalID=pgenetics&manu=PGENETICS-D-22-00123R2

**Press Queries**

---

## [Editor Report · Acceptance letter]

27 Aug 2022

PGENETICS-D-22-00123R2 

Repair of Mismatched Templates during Rad51-dependent Break-Induced Replication 

Dear Dr Haber, 

We are pleased to inform you that your manuscript entitled "Repair of Mismatched Templates during Rad51-dependent Break-Induced Replication" has been formally accepted for publication in PLOS Genetics! Your manuscript is now with our production department and you will be notified of the publication date in due course.

With kind regards,

Agnes Pap

PLOS Genetics

On behalf of:
